# On-the-Fly Test-time Adaptation for Medical Image Segmentation

**Jeya Maria Jose Valanarasu**[1]                                      JVALANA1@JH.EDU
[1] *Department of Electrical and Computer Engineering, Johns Hopkins University*
**Pengfei Guo**[2]                                                    PGUO4@JH.EDU
[2] *Department of Computer Science, Johns Hopkins University*
**Vibashan VS**[1]                                                   VVISHNU2@JH.EDU
**Vishal M. Patel**[1,2]                                             VPATEL36@JH.EDU

**Editors:** Accepted for publication at MIDL 2023

## Abstract

Adapting the source model to target data distribution at test-time is an efficient solution for the data-shift problem. Previous methods solve this by adapting the model to target distribution by using techniques like entropy minimization or regularization. In these methods, the models are still updated by back-propagation using an unsupervised loss on complete test data distribution. In real-world clinical settings, it makes more sense to adapt a model to a new test image on-the-fly and avoid model update during inference due to privacy concerns and lack of computing resource at deployment. To this end, we propose a new setting - On-the-Fly Adaptation which is zero-shot and episodic (*i.e.*, the model is adapted to a single image at a time and also does not perform any back-propagation during test-time). To achieve this, we propose a new framework called Adaptive UNet where each convolutional block is equipped with an adaptive batch normalization layer to adapt the features with respect to a domain code. The domain code is generated using a domain prior generator specially trained on medical images. During test-time, the model takes in just the new test image and generates a domain code to adapt the features of source model according to the test data instance. We validate the performance on both 2D and 3D data distribution shifts where we get a better performance compared to previous test-time adaptation methods while not performing back-propagation during test-time. The code can be found here: https://github.com/jeya-maria-jose/On-The-Fly-Adaptation
**Keywords:** Test-time adaptation, medical image segmentation.

## 1. Introduction

Image segmentation is a major task in medical imaging as it is essential for computer-aided diagnosis and image-guided surgery systems. In the past few years, deep learning-based solutions have been widely popular for medical image segmentation. Many convolutional methods (Ronneberger et al., 2015; Zhou et al., 2018; Milletari et al., 2016; Islam et al., 2018; Valanarasu et al., 2020) and transformer-based methods (Chen et al., 2021b; Valanarasu et al., 2021) have been proposed for various medical image segmentation tasks showing very good performance. However, a major problem with deep neural networks (DNN) is that they are highly dependent on the dataset that they are trained on. If a DNN is trained on a specific dataset and tested on a different dataset, the performance usually

Table 1: Comparison between different adaptation problem setups. Notation: source ($s$), target ($t$), data distribution $X$, label distribution $Y$.

| Setting | Source data | Target data | Train loss | Test loss |
|---|---|---|---|---|
| Source training | $X_s^{train}, Y_s^{train}$ | - | $\mathcal{L}_{det}(X_s^{train}, Y_s^{train})$ | - |
| Oracle | - | $X_t^{train}, Y_t^{train}$ | $\mathcal{L}_{det}(X_t^{train}, y_t^{train})$ | - |
| Unsupervised domain adaptation | $X_s^{train}, Y_s^{train}$ | $X_t^{train}$ | $\mathcal{L}_{det}(X_s^{train}, Y_s^{train}) + \mathcal{L}_{da}(X_s^{train}, X_t^{train})$ | - |
| Source free adaptation | - | $X_t^{train}$ | $\mathcal{L}_{da}(X_t^{train})$ | - |
| Test-time training | $X_s^{train}, Y_s^{train}$ | $X_t^{test}$ | $\mathcal{L}_{det}(X_s^{train}, Y_s^{train}) + \mathcal{L}_{aux}(X_s^{train})$ | $\mathcal{L}_{aux}(X_t^{test})$ |
| Fully test-time adaptation | - | $X_t^{test}$ | - | $\mathcal{L}_{tta}(X_t^{test})$ |
| On-the-Fly adaptation | - | $x_t^i \epsilon X_t^{test}$ | - | - |

drops even if they are of the same modality. This happens due to occurrence of many shifts like camera/scanner parameters, resolution, intensity, and contrast variations. This drop in performance makes DNN-based solutions for medical imaging tasks impractical to be adopted for real-time clinical use. For clinical use, the model needs to be robust as there can be small changes in the test data distribution even if they are of the same modality.

Many domain adaptation techniques for medical image segmentation have looked into solving this problem (Guan and Liu, 2021). However, this setting assumes that we have access to the source model, source data as well as the target data. Another setting very close to real-time is fully test-time adaptation (Wang et al., 2021; Karani et al., 2021) where we assume that we do not have access to the source data and adapt the model to the target data by performing one back propagation per sample. However, the model is adapted to the complete test distribution as the model weights are updated for at least one complete epoch. This setting can be considered one-shot adaptation as the model sees all the data in the distribution at least once. This can also be extended for few-shot adaptation to further adapt the model during test time. However, this setting is also not clinically deployable as we need a complete distribution to perform the adaptation and get the new model weights. Also, performing back-propagation during test-time means that we still have to do some training during the deployment-time although it is unsupervised. To get a clear understanding of these existing settings, let us assume $X$ and $Y$ represent the set of input data and labels, respectively. Let the source distribution be represented as $X_s$, $Y_s$ and the target distribution as $X_t$, $Y_t$. In normal source training, we train the model using $X_s^{train}$, $Y_s^{train}$ and test the model on $X_s^{test}$. All the previous settings are tabulated in Table 1.

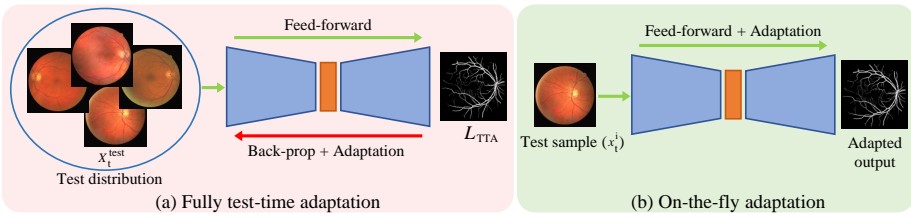

Figure 1: Comparison between (a) Test-time adaptation, and (b) On-the-Fly Adaptation.

In this work, we propose a clinically motivated setting called On-the-Fly test-time adaptation where the model adapts to a single image/volume at a time without any back-

propagation. Here, we just attempt to adapt our model to the new data instance. This makes On-the-Fly adaptation zero-shot as it does not really involve any gradient back-propagation during test-time. Also, as the model is reset for every data instance there is no need to assume the availability of the complete target distribution to perform adaptation since patient data may come with privacy concerns. This makes On-the-Fly adaptation episodic as it resets to original weight for adapting to each data-instance. On-the-Fly Adaptation is a more useful scenario in the current trend as there has been a shift of laboratory to bed-side settings for medical imaging (Vashist, 2017).

To solve this problem, we propose a new framework called Adaptive-UNet where the model is equipped with adaptive batch-norm layers in both the encoder and decoder to adapt to a select domain code. The domain code is generated using a domain prior generator. So, during test time, the model just takes in a single image/volume as input and feeds it to the encoder of the segmentation network as well as the domain prior generator. The domain code generated from the domain prior generator is used as a domain prior to adaptively normalize the features extracted from the segmentation encoder.

In summary, the following are the contributions of this work:

- We introduce On-the-Fly Adaptation which is more closer to real-world clinical scenarios where the adaptation is zero-shot and episodic removing the assumption of the availability of complete target distribution and back-propagation during test-phase.
- We propose Adaptive-UNet, a new framework that learns to adapt to a new test data instance making use of a domain code and adaptive batch normalization.
- We validate our method for 9 domain shifts in medical image segmentation for 2D fundus images and 3D MRI volumes where we get better performance than recent test-time adaptation methods.

## 2. Related Works

**Unsupervised Domain Adaptation** for medical image segmentation is a widely explored topic. Methods like feature alignment using adversarial training (Javanmardi and Tasdizen, 2018; Panfilov et al., 2019), disentangling the representation (Yang et al., 2019), ensembling and using soft labels (Perone et al., 2019) have been proposed. These methods, however, use the training distribution of both source and target data for adaptation which is not always feasible for medical imaging due to privacy concerns.

**Source-free Unsupervised Domain Adaptation** works for medical image segmentation assume no availability of source data. In (Bateson et al., 2020), a label-free entropy loss is defined over target distribution with a domain-invariant prior. In (Chen et al., 2021a), an uncertainty aware denoised pseudo label method is proposed.

**Test-time Adaptation** methods such as TENT (Wang et al., 2021) uses entropy minimization of batch norm statistics to adapt to a new target distribution. Recently, (Hu et al., 2021) proposed using new losses like regional nuclear norm and contour regularization to improve test-time performance for medical image segmentation. Self domain adapted networks (He et al., 2020) use auto-encoder based adaptors to rapidly adapt to a new task at test-time. (Karani et al., 2021) proposed a per-test-image adaptation method where they adapt the image so as to obtain plausible segmentation. DINSeg (Liu et al., 2021) uses dynamic instance normalization to perform dynamic style transfer in a learnable manner

proposing a plug and play method for increasing robustness. (Ouyang et al., 2022) proposed a causality inspired domain generalization method for medical image segmentation. Recently, masked auto-encoders were proposed to perform test-time training (Gandelsman et al., 2022). Note that all the previous methods perform back-propagation on network weights during test-time.

## 3. Method - Adaptive UNet

To solve On-the-Fly Adaptation, we propose an Adaptive UNet framework where we make use of adaptive batch normalization and a domain prior. The intuition behind our proposed Adaptive UNet framework is that we try to normalize the features with a prior during test-time. The source model is trained such that the features are normalized by a prior extracted from the current modality. So during test time, the prior of the new modality normalizes the features to handle the domain shift.

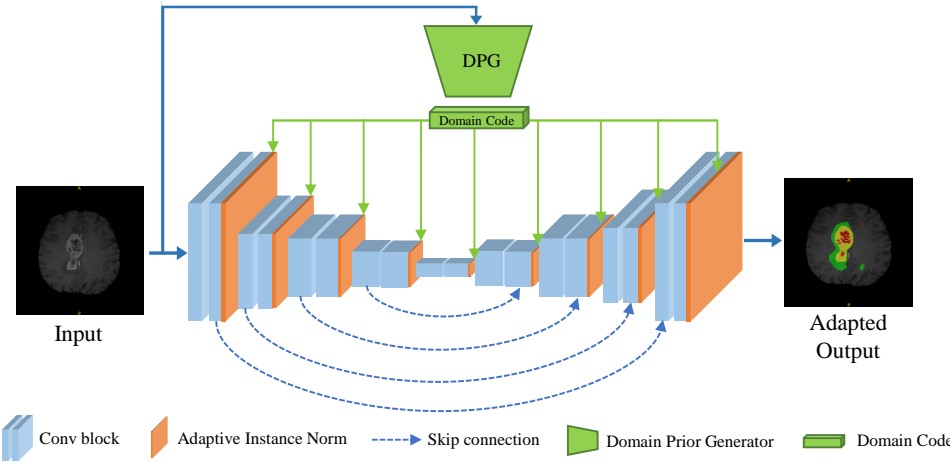

Figure 2: Overview of Adaptive UNet framework.

**Network Details:** We follow the skeleton of a generic UNet architecture (Ronneberger et al., 2015). We use 5 conv blocks in both the encoder and decoder, respectively. Each conv block in the encoder consists of a conv layer, adaptive batch normalization, ReLU activation and a max-pooling layer. Each conv block in the decoder consists of a conv layer, adaptive batch normalization, ReLU activation and an upsampling layer. For upsampling, we use bilinear interpolation. For our experiments on 3D volumes, we use a 3D UNet architecture (Çiçek et al., 2016) with the same setup replacing 2D conv layer with 3D conv layers, 2D max-pooling with 3D max-pooling and bilinear upsampling with trilinear upsampling.

**Adaptive Batch Normalization:** Batch Normalization (BN) layers (Ioffe and Szegedy, 2015) are used in DNNs to mitigate the issue of internal co-variate shifts. It normalizes the features in the network helping in training and faster convergence. BN is defined as:

$$z = \gamma \left( \frac{X - \mu(X)}{\sigma(X)} \right) + \beta, \tag{1}$$

where $x$ is the input batch, $z$ represents the output, $\mu$ represents the mean $E[X]$, $\sigma$ represents standard deviation $\sqrt{Var(X)}$. Here, $\gamma$ and $\beta$ are learnable parameters which control the scaling and shifting while normalizing.

Adaptive Instance normalization (AdaIN) (Huang and Belongie, 2017) is used to align the mean and standard deviation of two feature codes (usually one being context and another being style). AdaIN can be defined as:

$$f_z = \sigma(f_y) \left( \frac{f_x - \mu(f_x)}{\sigma(f_x)} \right) + \mu(f_y),$$

(2)

where $f_x$ and $f_y$ are the two feature codes, $f_z$ is the normalized output feature vector, $\mu$ represents the mean and $\sigma$ represents standard deviation.

In Adaptive UNet, we make use of Adaptive Batch Normalization (AdaBN) which basically learns scaling and shifting operation while adaptively normalizing the batch statistics between two codes. AdaBN can be defined as:

$$f_z = \gamma \left( \sigma(f_y) \left( \frac{f_x - \mu(f_x)}{\sigma(f_x)} \right) + \mu(f_y) \right) + \beta.$$

(3)

where $f_x$ and $f_y$ are two feature codes, $\mu$ represents the mean, $\sigma$ represents standard deviation, $\gamma$ and $\beta$ are the scaling and shifting parameters. Note that our formulation is a bit different from AdaBN as explained in (Li et al., 2016) as it tries to shift the model to test data's mean and standard deviation instead of aligning them. Here, we align the codes while also learning how to align them by learning the shift and scale parameters.

**Domain Prior Generator:** The Domain Prior Generator (DPG) is an encoder of a pre-trained auto-encoder. DPG is a pre-trained encoder that takes in an input image and outputs the domain code. We first pre-train a UNet as a denoising auto-encoder for medical images. This task is self-supervised as we just try to predict the original image while feeding an augmented version of the data as input. Doing this helps the model learn an abstract code in the latent space. We train the model on a variety of medical data consisting of different modalities. More details can be found in the appendix. We make sure that the distribution of data that we conduct experiments to validate Adaptive UNet do not overlap with the data that the auto-encoder is trained on. However, it does have the data of similar modality. This helps the encoder generate different domain codes for different modalities. For example, two images of T1 MRI would have their corresponding domain codes closer in the latent space when compared to T1 MRI and T2 MRI. The domain prior generated is a 1D feature vector from the latent space of the encoder. The mean and variance of this feature vector is used to normalize the original features.

**On-the-fly adaptation:** During the training phase of source model, we feed in the input image $X$ to both the encoder of UNet and the pre-trained domain prior generator. The domain code obtained from the domain prior generator $f_y$ is passed to the AdaBN layers in the encoder and decoder. The feature maps at the segmentation encoder $f_y$ are normalized according to the domain code using AdaBN. So, in the training itself the model has learned to adapt to the domain code of the current modality/distribution. The learnable parameters $\gamma$ and $\beta$ of AdaBN layers learn the scale and shift necessary to adapt to the style code at each level to provide the optimal segmentation prediction. Note that the weights are updated only for the UNet segmentation network. The pre-trained domain prior generator is frozen during training the source model.

During inference, a model trained on source domain is validated on a target data instance $x_t$, we forward the image $X$ to both the domain prior generator and the source model. First, we generate the domain code $f_y$ using the domain prior generator for the new image. Next we pass this domain code to all the AdaBN layers in Adaptive UNet. Thus the features

extracted at each layer $f_x$ of Adaptive UNet are adapted to the new domain code $f_y$. So, the model is thus adapted according to the code of the new modality/target domain. There is no back-propagation involved as the features are adapted in feed-forward itself. Also, as the model weights are not changed, this framework is episodic and does not depend on the entire test data distribution for validation. An overview of the proposed framework is illustrated in Fig. 2.

## 4. Experiments and Results

**Datasets:** For 2D experiments, we focus on the task of retinal vessel segmentation from fundus images. We make use of the following datasets: CHASE (Fraz et al., 2012), RITE (Hu et al., 2013) and HRF (Odstrčilík et al., 2009). CHASE contains 28 retina images with a resolution of 999×960 collected from 14 school children with a hand-held Nidek NM-200-D fundus camera. RITE consists of 40 images of resolution 768×584 collected from people aging from 25 to 90 using a Canon CR5 non-mydriatic 3CCD camera. HRF contains 18 images collected from 18 human subjects using a Canon CR-1 fundus camera of around resolution 3504×2336. There exists a domain shift among these datasets as they vary with respect to camera properties, age of patients and resolution etc. The datasets are separated into a randomized 80-20 split wherever test split was not given.

For 3D experiments, we focus on brain tumor segmentation from MRI volumes. We make use of the BraTS 2019 dataset (Menze et al., 2014; Bakas et al., 2017, 2018) which consists of four modalities- FLAIR, T1, T1ce and T2. We study the domain shift problems between these four modalities for volumetric segmentation of brain tumor. This is a multi-class segmentation problem with 4 labels. We randomly split the dataset into 266 for training and 69 for validation. We do this as the ground truth is not provided publicly for the original validation dataset.

For pre-training DPG on MRI images, we make use of Kaggle MRI dataset and IXI dataset. For the fundus experiments, we make use of the fundus1000 dataset. More details can be found in the appendix.

**Implementation Details:** We use Pytorch framework for implementing Adaptive UNet. For 2D experiments, we use a combination of binary cross entropy (BCE) and dice loss to train Adaptive UNet. The loss between the prediction $\hat{y}$ and the target $y$ is formulated as:
$$\mathcal{L} = \lambda BCE(\hat{y}, y) + Dice(\hat{y}, y). \tag{4}$$
We use an Adam optimizer with a learning rate of 0.0001 and momentum of 0.9. We also use a cosine annealing learning rate scheduler with a minimum learning rate upto 0.00001. The batch size is set equal to 8. For 3D experiments, we use a similar loss but use a learning rate of 0.001 while also reducing the batch size to 2. For baselines, we did test the other methods with different hyper-parameters like learning rate to get the best results. DPG trained on crops of 256 x 256 with an Adam optimizer with a loss of 0.0005 and batch size of 4. It is trained on a simple L1 loss for 300 epochs.

**Performance Comparison:** We compare our proposed method with recent test-time adaptation methods like TENT (Wang et al., 2021), (Hu et al., 2021) (RN+CR loss) , self domain adapted network (SDA) (He et al., 2020), and (Karani et al., 2021). In Table 2, we present the results of 2D experiments for 6 different domain shifts in fundus image. In Table 3, we present the results of 3D experiments for 3 different domain shifts in MRI

Table 2: Results for 2D Domain shifts. Numbers correspond to dice score.

| Type | Method | CHASE -> HRF | CHASE -> RITE | HRF -> CHASE | HRF -> RITE | RITE -> CHASE | RITE -> HRF |
|---|---|---|---|---|---|---|---|
| Source-Training | Direct Testing | 61.95 | 70.88 | 31.65 | 62.87 | 44.65 | 59.67 |
| Test-Time Adaptation (One-shot) | SDA (He et al., 2020) | 11.19 | 6.22 | 3.64 | 4.75 | 0.91 | 5.15 |
| | TENT (Wang et al., 2021) | 61.20 | 73.65 | 19.20 | 58.64 | 63.76 | 57.14 |
| | RN+CR (Hu et al., 2021) | 61.31 | 72.85 | 21.20 | 57.35 | 63.95 | 59.13 |
| | (Karani et al., 2021) | 68.92 | 73.58 | 41.99 | 60.32 | 64.91 | 61.51 |
| Test-Time Adaptation (Ten-shot) | SDA (He et al., 2020) | 3.11 | 5.13 | 1.84 | 5.48 | 7.33 | 5.15 |
| | TENT (Wang et al., 2021) | 61.17 | 73.62 | 19.22 | 58.64 | 63.72 | 57.10 |
| | RN+CR (Hu et al., 2021) | 61.42 | 72.88 | 21.22 | 57.32 | 63.91 | 59.15 |
| | (Karani et al., 2021) | 69.07 | 73.88 | 42.35 | **60.98** | 64.56 | 62.14 |
| **On-the-Fly Adaptation (Zero-shot)** | **Adaptive UNet (Ours)** | **70.19** | **74.27** | **50.27** | 59.59 | **65.98** | **63.14** |
| Target-Training | Oracle | 76.88 | 77.69 | 75.78 | 77.69 | 75.78 | 76.88 |

modality. In both the tables, the first row corresponds to the direct adaptation results where we train the model on source domain and report the results of those models while testing on the target domain without any adaptation. The last row corresponds to the oracle which is the maximum possible performance when the model is trained on the target train distribution and tested on the target test distribution. Note that the 3D experiments have two target-training configurations- Uni-modal and Multi-modal. Uni-modal oracle corresponds to the configuration where we only use one modality and multi-modal oracle corresponds to the case where we use all four modalities to train the model. The compared test-time adaptation methods are presented in both one-shot and ten-shot settings. In one-shot setting, the model weights are adapted by back-propagation for one epoch using the test distribution. In ten-shot setting, the model weights are adapted for ten epochs by back-propagation using the test distribution. Previous test-time papers for segmentation (Wang et al., 2021) show results only for 1 and 10 epochs. The reason is that training a network for a large number of epochs corresponds to test-time-training and not test-time-adaptation. However, training the network for 100 epochs using these methods do not really change the results much. It can be noted that all the previous methods perform back-propagation during test-time while our method does not.

From Tables 2 and 3, it can be inferred that there is a considerable drop in performance while directly testing the source model on the target domain. We repeat experiments thrice and report mean and variance. This drop is expected as there exits a domain shift between the source and the target distribution. SDA does not perform well, especially in relatively small datasets (Table 2). Since SDA requires training a set of auto-encoders to provide supervision during test-time adaption, only training on a small amount of data may result in overfitting and consequently lower adaptation performance. TENT, RN+CR, and (Karani et al., 2021) methods improve the performance in most cases as they try to reduce the entropy and regularize the batch-norm statistics for the target distribution. Our proposed method shows a considerable improvement over the direct testing as well as test-time baselines on almost all domain shifts achieving state-of-the-art adaptation performance. Note that brain tumor segmentation from 3D volumes is a multi-class segmentation problem which shows that our method can be successfully adopted multi-class problems as well.

Table 3: Results for 3D Domain shifts. Numbers correspond to dice score reported in the following order: WT/TC/ET. WT = Whole Tumor, TC = Tumor Core, ET = Enhancing Tumor.

| Type | Method | T1 ->T1ce | T1ce ->T1 | FLAIR ->T1ce |
|---|---|---|---|---|
| Source-Training | Direct Testing | 48.74/52.35/36.48 | 59.25/33.78/9.02 | 24.35/39.51/29.38 |
| Test-Time Adaptation (One-shot) | SDA (He et al., 2020) | 10.89/48.98/29.47 | 12.03/6.52/2.25 | 13.37/16.72/7.62 |
| | TENT (Wang et al., 2021) | 57.41/55.31/39.79 | 68.25/46.99/6.61 | 24.33/41.95/22.90 |
| | RN+CR (Hu et al., 2021) | 55.21/54.62/38.21 | 67.56/47.21/5.52 | 24.31/41.85/22.82 |
| | (Karani et al., 2021) | 58.65/57.21/38.49 | 67.82/47.67/6.00 | 24.46/41.90/29.61 |
| Test-Time Adaptation (Ten-shot) | SDA (He et al., 2020) | 14.21/52.65/32.34 | 8.91/2.96/10.82 | 14.01/28.11/13.88 |
| | TENT (Wang et al., 2021) | 55.47/52.89/39.17 | 66.23/42.52/**12.14** | 19.96/36.25/22.50 |
| | RN+CR (Hu et al., 2021) | 57.30/54.68/**40.01** | 60.82/42.87/12.11 | 19.39/33.96/17.09 |
| | (Karani et al., 2021) | 58.94/57.61/38.67 | **68.01**/47.94/6.02 | 24.55/41.99/**29.82** |
| On-the-Fly Adaptation (Zero-shot) | Adaptive UNet (Ours) | **60.66/58.73**/39.30 | 65.08/**48.09**/8.78 | **24.86/42.27**/29.20 |
| Target-Training | Uni-Modal Oracle | 73.49/74.54/67.97 | 70.88/56.60/26.70 | 73.49/74.54/67.97 |
| | Multi-Modal Oracle | 91.06/70.09/78.97 | | |

We also provide sample qualitative results in Fig. 3. It can be observed that without any adaptation, the segmentation predictions are noisy and contain over-segmentation. While recent test-time methods improve the prediction, they still suffer from mis-classification of pixels (see MRI predictions in Fig. 3 for TENT/RN+CR) and also over-segmentation (see fundus predictions in Fig. 3 for TENT/RN+CR). Our method achieves good segmentation prediction that is very close to the oracle prediction and the ground-truth.

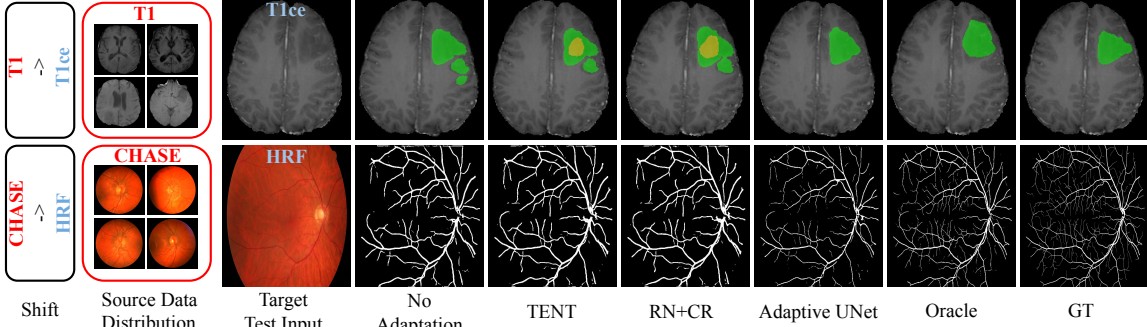

Figure 3: Qualitative Results. Yellow regions corresponds to non-enhancing tumor while the green regions correspond to edema.

**Discussion:** From our experiments, we find that our method works pretty well for 2D experiments when compared to 3D experiments. This observation can be understood as the 2D experiments consider domain shifts within the same modality with differences in camera/sensor properties and type of patient. The 3D experiments consider cross-modality domain shifts where the MRI sequences are themselves different. This is a more difficult task as each sequence extracts different types of features. However, we get a considerable boost over other test-time methods even though our setting is episodic and zero-shot.

## 5. Conclusion

In this work, we propose a new adaptation setting called On-the-Fly adaptation. In this setting, the adaptation is episodic and zero-shot thus assuming no availability of the entire target distribution and model update during test-time. We propose a new framework-Adaptive UNet to solve this adaptation problem by making using of adaptive batch normalization and domain priors. We validate our model on both 2D and 3D domain shifts and show that the proposed method achieves a competitive performance.

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

## Appendix A. More details on Domain Prior Generator:

Table 4 shows the data on which DPG is trained on. For the MRI experiments, we pre-train DPG on MRI images from Kaggle MRI dataset and IXI dataset. Note that DPG is a 2D architecture. So, we only use 2D images to train the network. Even for 3D volumetric experiments, we still use a 2D DPG model to extract the prior. We sample a slice from the input MRI volume (the center slice) and feed it to the DPG to extract domain prior. For 2D experiments, DPG is pre-trained on fundus images from Fundus. Note that while training we add some noise to the input and try to reconstruct the original input similar to a denoising auto-encoder. The domain prior generator is trained just for reconstruction using a set of medical images. This can be considered as a self-supervised pre-training setup where the model is trained on images shown in Table 4 with no labels. From this, we can understand that DPG is just another pre-trained network and does not have any knowledge about the data that we used to train the source models in our main experiments.

Table 4: Details on the pre-training dataset.

| Type | Dataset | Resolution | Number of data | Information |
|---|---|---|---|---|
| 2D Experiments | fundus1000 | ~3000x2500 | 1000 | 39 characteristics |
| 3D Experiments | Kaggle MRI | 256x256 | 98 | T1/T2/T1ce/FLAIR |
| | IXI | 256x256 | 600 | T1/T2/PD/MRA/DTI |

Table 5: Limitations. Numbers correspond to dice score reported in the following order: WT/TC/ET. WT = Whole Tumor, TC = Tumor Core, ET = Enhancing Tumor.

| Type | Method | T1 –>T2 | T1 –>FLAIR | FLAIR –>T1 |
|---|---|---|---|---|
| Source-Training | Direct Testing | 10.58/14.61/7.81 | 13.32/34.57/19.90 | 10.84/20.02/23.71 |
| Test-Time Adaptation (One-shot) | SDA (He et al., 2020) | 3.15/2.16/1.94 | 2.03/1.80/1.55 | 3.37/2.84/1.61 |
| | TENT (Wang et al., 2021) | 8.27/8.40/4.37 | 12.94/26.79/13.64 | 11.13/19.83/5.03 |
| | RN+CR (Hu et al., 2021) | 8.10/8.21/4.78 | 12.54/25.55/13.19 | 10.64/18.88/4.76 |
| | (Karani et al., 2021) | 8.59/9.00/5.18 | 15.88/29.51/13.55 | 10.44/19.50/4.84 |
| Test-Time Adaptation (Ten-shot) | SDA (He et al., 2020) | 3.18/2.24/2.05 | 2.18/2.01/1.88 | 3.11/2.80/1.52 |
| | TENT (Wang et al., 2021) | 8.39/8.48/4.30 | 13.10/27.02/13.60 | 11.8/19.55/4.94 |
| | RN+CR (Hu et al., 2021) | 8.11/8.25/4.75 | 12.68/25.71/13.18 | 10.10/18.91/4.57 |
| | (Karani et al., 2021) | 8.62/8.92/5.24 | 15.59/29.56/11.52 | 10.58/19.58/4.88 |
| On-the-Fly Adaptation (Zero-shot) | Adaptive UNet (Ours) | 6.54/5.02/2.72 | 9.01/17.72/8.22 | 8.47/14.69/6.09 |
| Target-Training | Uni-Modal Oracle | 83.21/33.73/62.08 | 88.19/31.45/61.70 | 72.54/28.77/58.37 |
| | Multi-Modal Oracle | 91.06/70.09/78.97 | | |

## Appendix B. Limitations:

If the domain shift is huge, the no adaptation performance is itself very bad (around 10 dice). This causes the boost in performance to be negligible for all methods including ours as seen in Table 5. Addressing a huge domain shift is still a limitation and we leave it for future work.

## Appendix C. Future Works:

It is worthy to note that our method does use a domain prior to adapt the model for the test data instance. However, the domain prior generator does not use any data from the source or target data distribution during its pre-training. It is simply trained on publicly available medical images of various modalities. In this work, we used different DPGs for 2D and 3D experiments because the type of data we trained for 2D (fundus) was hugely different from that of 3D (MRI). We did perform experiments where we tried to have a common DPG for both 2D and 3D but the performance was not better than having separate DPG for 2D and 3D. The data which we pre-train DPG can further be improved by incorporation a large collection of public medical datasets. We leave the topic of coming up with a better and stronger DPG for future research. In this work, we show that a simple DPG trained on a certain number of public data is itself adequate to perform on-the-fly adaptation for medical image segmentation.

