# OpenReview forum: "On-the-Fly Test-time Adaptation for Medical Image Segmentation"
_MIDL.io/2023/Conference — MIDL 2023 Poster_

### Official Review · Reviewer_Pr6a · 2023-02-01

**Confidence:** 4
**Preliminary Rating:** 2

**Summary:**

An online test-time adaptation method is developed for medical image segmentation. The authors use a pre-trained domain prior genitor to adaptively obtain the parameters in the batch normalization layers, so that the model does not need back-propagations to update the parameters.  Experiments was conducted on both 2D and 3D datasets to show its effectiveness.

**Strengths:**

1, the authors consider a new setting of test-time adaptation: applying a pre-trained model to a target domain directly without using backpropagation to update its parameters.

2, The proposed method was validated on both 2D and 3D datasets for experiments.


**Weaknesses:**

1, Some part of the method does not seem to be rational, especially the domain prior generator. The authors used an anto-encoder to capture the “domain prior”. However, the latent code in auto-encoder mainly represents the input image’s contextual information, so that the decoder can reconstruct the input image well. The authors did not use any regularizations to force the encoder capture “domain prior”. Therefore, I’m not convinced that the author-encoder can capture domain prior.

2, Some part of the method is not clear. For example, how was the “domain code” is mapped to the parameters in the AdaBN layer is not described. In eq. 3, how was gamma and beta obtained?

3, The experimental comparison seems to be unfair. The compared method TENT and RN+CR use the target domain images for backpropagation till converge. However, the authors only updated the mode for one or ten epochs in the target domain.


**Deanonymize Review:**

no

**Detailed Comments:**

1, The implementation is not clear enough. How was the auto-encoder trained? It seems that the auto-encoder need to be trained with a large dataset with different modalities. But its implementation was not mentioned. What was the patch size for training/inference?

2, There is not ablation study for the method. The DPG seems to be important for achieving the good performance. The authors should compare pretraining DPG using only the source domain data with pre-training it with the large-scale data including different modalities.

3, The concept of “source-free unsupervised domain adaptation” and “test-time adaptation” are the same thing in my understanding. Both of them refers to adapting a pre-trained model to a new target domain without access to the source domain images and annotations in the target domain. However, the authors listed related works according to these two terms in the related work. Please consider to merge them together, or clarify the difference in the text.


**Paper Type:**

methodological development

**Questions To Address In The Rebuttal:**

1, Please provide the rational for using auto-encoder to capture the “domain prior”. Why does it obtain domain information rather than the textual information.

2, Why not applying TENT and RN+CR with convergence? The results with more epochs need to be compared.

3, The authors split the images into training and validation. Why not using a testing dataset for evaluation? The validation data is often used to tune hyper-parameters.

4, How the domain code is connected with the BN layer must be clarified.

---

### Official Review · Reviewer_U81j · 2023-02-02

**Confidence:** 4
**Preliminary Rating:** 4

**Summary:**

The paper describe a method to adapt a model for domain shift at test time for a single image.  The method does not require updates to the original network weights via back propagation, but rather makes use of a "domain prior generator" network which generates a so-called domain code, used to adaptively normalize the batch statistics.


**Strengths:**

The paper is generally well written and in clear, if slightly imperfect, English.  The figures are sufficient.  The problem tackled is an interesting one and is demonstrated in both 2D and 3D and compared to the state of the art in the field on the same data.

**Weaknesses:**

The paper gives the impression that the authors struggled to meet the page limit and important information has been moved to the Appendix (e.g. regarding training of the domain prior generator).  It should be possible to tighten up the text and fit the important details into the main body of the paper.
I do not have a good sense of what is needed to implement this method in the end - e.g. what kind of data I would need to have to train the domain prior generator.
The text should also be revised for minor typos and language improvements e.g. "trained on a bunch of images" or "our method works pretty well for 2D experiments" could be rephrased to be more precise/scientific and there are some areas where correctness of the English could be improved.

**Deanonymize Review:**

no

**Detailed Comments:**

The information about the training of the DPG should be in the main part of the paper since this is crucial to the method working.  If DPG is only trained with fundusimage1000 then it has only seen one "domain" of fundus images, albeit with augmentations, so how is it able to generate good domain codes for images from several new domains (different cameras, resolutions etc)?    Does the Kaggle MRI set have FLAIR, T1, T1ce, T2 (this should be specified in the text), or otherwise I have a similar question -  how does the DPG learn to generate good domain codes for these types of images.  It is stated on page 5 that the data used for training the autoencoder (DPG) should have "data of a similar modality" as that which will be used to conduct experiments.   The authors need to guide the reader as to what is meant by that and what kind of variability is required in the training set of the DPG.


Table 4 is entirely uninformative and should be properly filled with information about the numbers/types/resolutions of images in training/validation sets.

Please check that image captions are sufficiently informative.  E.g. what is the yellow colour on the segmentations in Figure 3?

In section 1 please use the term "domain prior generator" to describe the "pre-trained encoder" so that terminology conventions are the same throughout.

The word "oracle" is used in multiple places/tables starting in Table 1, but is only defined once, in passing, at the end of page 6 as far as I can see.  It should be defined early on in the text/captions.

The Table numbers in the Appendix are not referred to correctly in the text.



**Paper Type:**

methodological development

**Questions To Address In The Rebuttal:**

I would like the authors to address all points mentioned in the detailed comments above, as well as revise the text for minor language improvements.  The method is interesting but I do not understand well what is needed to make a DPG that will function well for new domains.

---

### Official Review · Reviewer_iSbY · 2023-02-02

**Confidence:** 4
**Preliminary Rating:** 4

**Summary:**

The authors propose a method for domain adaption. The method involves applying a pre-trained encoder to an unseen test-time image, and feeding the latent encoding to adaptive batchnorm layers in the segmentation network, enabling adaption without performing any backpropagation on the test-time image. The authors find favourable performance compared to methods that adapt to the test-time image.

UPDATE: I've updated to weak accept. I'm still not 100 convinced by the comparisons - the baseline and proposed methods have access to different datasets which them hard to compare - but I think there is enough novelty here that the community will find the work interesting.

**Strengths:**

- The paper is generally clear and easy to follow
- The authors evaluate on both 2D and 3D medical datasetes
- The method generally outperform the test-time adaption methods compared against
- Not needing to perform any backpropagation on the test image is an advantage

**Weaknesses:**

I worry the comparisons performed are not fair. The baseline methods have access to both the source and target data. The proposed method has access to the source, target data, and an additional unlabelled dataset used for training the domain encoder. These additional datasets are substantially larger than the source datasets. A fair comparison should involve methods that incorporate the large unlabelled additional datasets during training, perhaps through some unsupervised training.

Table 5 seeks to demonstrate that none of the methods perform will in the performance of large domain shifts, but the table leaves out several of the baseline methods. I would expect some of the test-time adaptation methods to perform better on large domain shifts, especially the ones trained for ten epochs. These should be included in this comparison and, if some of the TTA methods do perform substantially better than the proposed method there, this needs to be more prominently discussed as a weakness of the proposed method.

**Deanonymize Review:**

no

**Detailed Comments:**

Fail to highlight the best result in the T1ce –>T1 comparison (TENT)

In figure 3, the proposed method looks closer to the GT than the oracle does, which is surprising - any explanation for this?

I'm unsure what the part in parentheses means in this sentence in limitations: "If the domain shift is huge, the no adaptation performance is itself very bad ( 10 dice).

**Paper Type:**

methodological development

**Questions To Address In The Rebuttal:**

The authors need to discuss the comparison. I'm not convinced the current experiments represent a fair comparison, as the proposed method has access to a larger dataset that the other methods do not.

I'd like to understand why the authors left out some of the methods when evaluating with large modality shifts, and see what the results look like with those methods included.

---

### Meta-Review · Area_Chair_cKrX · 2023-02-22

**Recommendation:** Accept (Poster)
**Confidence:** 5

**Metareview:**

On-the-Fly Test-time Adaptation for Medical Image Segmentation

The consensus is on weak acceptance, ranking from 2 acceptances and 1 mild rejecton. The contribution is on domain adaptation by using an adaptive batchnorms on the latent reprensetation used from a pre-trained encoder. This avoids retraining. Concerns are on possible unfair comparison and prior domain generation. Given the global ranking of submissions, the recommendation is leaning towards acceptance.

Recommendation towards Acceptance.